# Efficient Multi-modal Large Language Models via Visual Token Grouping

## Abstract

The development of Multi-modal Large Language Models (MLLMs) has significantly advanced various downstream applications, including visual question answering and image captioning. However, the substantial computational costs associated with processing high-resolution images and videos pose a barrier to their broader adoption. To address this challenge, compressing vision tokens in MLLMs has emerged as a promising approach to reduce inference costs. In this paper, we introduce `VisToG`, a novel grouping mechanism that leverages the capabilities of pretrained vision encoders to group similar image segments without the need for segmentation masks. With the isolated attention we adopt, `VisToG` can identify and eliminate redundant visual tokens, which effectively reduces computational demands. Extensive experiments demonstrate that the effectiveness of `VisToG`, maintaining 98.1% of the original performance while achieving a reduction of over 27% inference time.

## 1 Introduction

The advent of Large Language Models (LLMs) (Bai et al., 2023; Touvron et al., 2023; Achiam et al., 2023; Zhu et al., 2023), has revolutionized the landscape of natural language processing (NLP), enabling unprecedented advancements in tasks ranging from text generation to machine translation. With the need to incorporate information from other modalities such as vision, research on Multi-modal Large Language Models (MLLMs) Liu et al. (2024); Bai et al. (2023); Zhu et al. (2023) has attracted much attention. MLLMs emerged as a powerful paradigm, combining the strengths of both visual and textual modalities to achieve superior performance in tasks requiring cross-modal understanding However, the remarkable capabilities of these models come with substantial computational costs, particularly during the inference phase. This computational burden is exacerbated when encountering multi-modal inputs leading to a long input sequence, such as high-resolution images or videos. This limits their practical deployment in resource-constrained environments.

Typically, LLM costs the most for the MLLM computation because of the model size difference compared with the visual encoder and visual connector. For example, the widely used ViT-L Radford et al. (2021) only has 0.3B parameters while the language encoder typically has 7B or 13B parameters Chiang et al. (2023). Therefore, towards building an effective MLLM, current works focus on reducing the image tokens fed to the LLMs. Various approaches have been applied to serve this purpose. Main stream method includes training-free and finetuning. Training-free methods typically utilize the off-the-shelf pre-trained MLLMs and prune the visual tokens according to the attention in the transformer layers in LLMs Chen et al. (2024). While training-free method is plug-and-play, their performance are far from satisfying and they are often ineffective when applied to training because the in-training attention scores are unstable. Finetuning methods perform visual token reduction via operations on the image feature produced by the visual encoder, such as Adaptive Average Pooling Yao et al. (2024), convolution block or deformable attention block as visual abstractor Cha et al. (2024). However, these methods all conduct visual token reduction on the image features after feeding into the visual encoder, while leaving the potential of the pre-trained visual encoder not fully explored.

We observe that randomly chosen image tokens can serve as a strong baseline, indicating the redundancy of the image tokens. Further, if we deliberately modify the image tokens distribution following prior knowledge from humans, the performance can be greatly improved. Specifically,

the sampled tokens covered all semantic segments of the image. Motivated by this, we propose a novel visual token grouping mechanism aimed at reducing the inference costs of MLLMs by exploring the potential of the pre-trained vision encoder to reduce the redundant token while covering all semantic groups. Our approach leverages the inherent structure and redundancy present in visual data to condense the visual token representation with minimally sacrificing performance. By intelligently grouping visual tokens, we can significantly decrease the number of tokens processed by the model, thereby reducing computational overhead and accelerating inference times.

## 2 RELATED WORKS

### 2.1 MULTI-MODAL LARGE LANGUAGE MODELS

The success of Large Language Models (LLMs) has advanced various applications in the field of natural language processing. Its strong instruction-following ability and generalization power across tasks drive the researchers to build a multi-modal counterpart. GPT-4V from OpenAI has proven the potential of how an MLLM can do Yang et al. (2023). Researchers have put efforts to reimplement MLLMs similar to GPT-4V. The core design lies in how to connect the pre-trained visual encoder and the LLM. Resampler Bai et al. (2023) and Q-Former Li et al. (2023a); Dai et al. (2023); Zhang et al. (2024) employs learnable queries to represent visual tokens and force extracting the most relevant visual information from visual features by cross-attention layers. These works helped connect the visual and text modalities and achieved remarkable progress. To make the alignment between visual encoders and LLMs more effective and efficient, LLaVA Liu et al. (2024) uses a single projection layer to conduct the alignment. With a meticulously curated instruction-tuning dataset, it achieves remarkable performance, rivaling models trained on extremely large-scale datasets, yet it requires only a manageable training cost. The reduction in substantial training expenses greatly benefits the MLLM community.

### 2.2 EFFICIENT INFERENCE FOR LLM/MLLM

The auto-regressive nature of LLMs poses a great challenge on the deployment of LLMs. The quadratic complexity of computing the attention makes the generation process becomes much slower when the input token is longer. StreamingLLM Xiao et al. (2023) and FastGen Ge et al. (2023) prune the redundant attention computation to simplify the computation. Despite their success, they are designed for the single-modal LLM and are not proven successful when it comes to scenarios involving tokens from other modalities. For improving the efficiency of MLLMs, various works Li et al. (2023a); Zhu et al. (2023); Zhang et al. (2024) adopt Q-Former to map the images to fixed-length tokens. In the meantime, many works try to train smaller MLLMs with smaller backbone Yuan et al. (2023); Zhou et al. (2024) to handle the scenario with less computational resources, MoE-LLaVA Lin et al. (2024) incorporates a Mixture of Experts to address model sparsity, enhancing efficiency and performance. Another line of research tries to reduce the number of visual tokens while keeping the backbone of LLM unchanged. Since LLMs contribute most to the computation, the number of input tokens to the LLM becomes the bottleneck of the inference cost. Recently, DeCo Yao et al. (2024) employs 2D adaptive average pooling to down-sample the visual tokens at the patch level. Honeybee Cha et al. (2024) proposes to use ResNet Block and deformable attention to conduct the abstraction of the vision tokens. VoCo-LLaMA Ye et al. (2024) compress the vision tokens using the LLMs and leverage the attention distillation to let LLMs restore information from the specially defined VoCo tokens instead of the whole image tokens.

## 3 METHOD

In this section, we first recap how a typical MLLM works and then introduce `VisToG`, an innovative approach for efficient visual token grouping in MLLMs. `VisToG`introduces a novel grouping mechanism that leverages off-the-shelf pre-trained Vision Transformers to cluster similar image segments into semantically related concepts. By doing so, it effectively eliminates the need to encode redundant vision tokens, thereby optimizing computational efficiency.

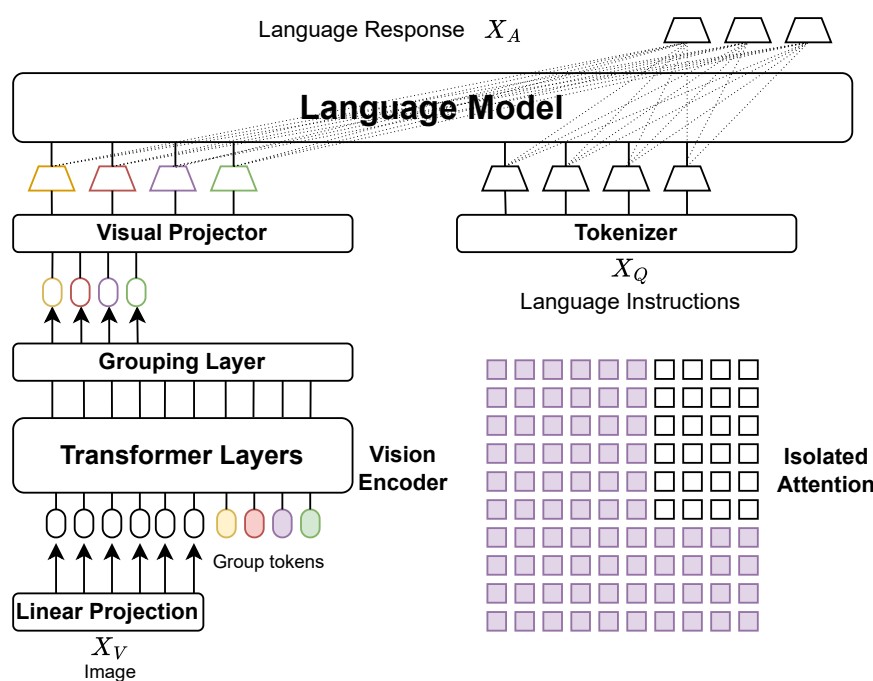

Figure 1: Overview of of our proposed `VisToG`. Semantic tokens are concatenated with the image patch tokens after linear projection and fed into the pre-trained vision encoder. Before the visual projector to LLM, a grouping layer is applied to group similar image segment tokens into semantically abstraction tokens of image. Besides, isolated attention is applied to ensure a better abstraction.

### 3.1 RECAP OF MULTI-MODAL LARGE LANGUAGE MODELS

MLLMs aim to develop a powerful model capable of generating responses that follow the instructions given for multi-modal inputs, including visual and textual data. MLLMs are typically composed of three core components: 1) Visual Encoder $E_v$: it converts an input image $I \in \mathbb{R}^{H \times W \times 3}$ into a set of distinctive visual embeddings $I_v \in \mathbb{R}^{N \times C}$. CLIP-ViT-L/14 with patch size 14 are widely adopted as the visual encoder due to their language-aligned pretrained nature, $N = HW/P^2$ denotes the number of visual embeddings. 2) Visual Projector $E_p$, parametrized by $W$, which is typically a multi-layer perceptron: this component translates visual embedding $I_v$ into the visual token $T_v$ in the textual embedding space $T$ with an appropriate dimension for the subsequent language model. This part substantially serves as a tokenizer for the input image 3) Large Lanugae Model$E_l$, parameterized by $\Phi$: it takes in both visual token $T_v$ and textual token $T_t$, and produces an appropriate response auto-regressively. For a sequence of response with length $L$, we compute the probability of the target answers $T_a$ by $T_a$ can be calculated by:

$$p(T_a|T_v, T_t) = \prod_{i=1}^{L} p_\phi(t_i|T_v, T_{t,<i}, T_{a,<i}), \tag{1}$$

In this framework, the computation burden lies in the LLM with many more parameters than the other components. Therefore, the number of input tokens influences most to the overall efficiency. Therefore, compressing visual tokens becomes the most popular approach to build an efficient MLLM.

## 3.2 GROUPING LAYER

To reduce the number of image tokens sent to the LLMs while minimizing the performance loss. We propose to utilize the power of the pre-trained Vision Encoder, motivated by the design of Xu et al. (2022), we add several learnable semantic tokens $\{\mathbf{Sem}_i\}_{i=1}^{N}$ before the transformer layer of ViT and concat it with the image segments $\{\mathbf{Img}_i\}_{i=1}^{M}$ after the linear projection, where $N$ and $M$ represents the number of semantic groups and original image segments tokens. ... The Grouping Block takes the learned semantic tokens and image segments tokens as input and merges all the segment tokens that are assigned to the same semantic components into a single new image segment, based on similarity in the embedding space. To be more specific, we compute the similarity matrix $\mathbf{A}$ between the semantic tokens $\{\mathbf{Sem}_i\}_{i=1}^{N}$ and image segments tokens $\{\mathbf{Img}_i\}_{i=1}^{M}$ through a Gumbel-Softmax operation computed over semantic tokens as

$$\mathbf{A}_{i,j} = \frac{\exp(W_q\mathbf{Sem}_i \cdot W_k\mathbf{Img}_j + \gamma_i)}{\sum\limits_{k=1}^{N} \exp(W_q\mathbf{Sem}_k \cdot W_k\mathbf{Img}_j + \gamma_k)} \tag{2}$$

where $W_q$ and $W_k$ are the weights of the learned linear projections for the semantic tokens and image segment tokens respectively and $\{\gamma_i\}$ are i.i.d random samples drawn from the Gumbel (0,1) distribution. Afterwards, we compute the semantic group to assign image segment tokens to by taking the one-hot operation of its argmax over all groups. Since the one-hot assignment operation via argmax is non-differentiable, we adopt the straight-through trick in Van Den Oord et al. (2017) to compute the assignment matrix as

$$\hat{\mathbf{A}} = \texttt{one-hot}(\mathbf{A}_{\text{argmax}}) + \mathbf{A} - \texttt{sg}(\mathbf{A}) \tag{3}$$

where $\texttt{sg}$ is the stop gradient operator. After assigning the image segment tokens to different semantic groups, we merge the embedding of all the tokens belonging to the same semantic group to form a new image token. For each new image token $\mathbf{VIS}_i$, it is a weighted sum of the image segment tokens assigned to a semantic group, which is computed as

$$\mathbf{VIS}_i = \mathbf{Sem}_i + W_o \frac{\sum\limits_{j=1}^{M} \hat{\mathbf{A}}_{i,j} W_v \mathbf{Img}_j}{\sum\limits_{j=1}^{M} \hat{\mathbf{A}}_{i,j} W_v} \tag{4}$$

where $W_v$ and $W_o$ are the learned weights to project the merged features. The structure of the grouping layer is illustrated in Fig. 2

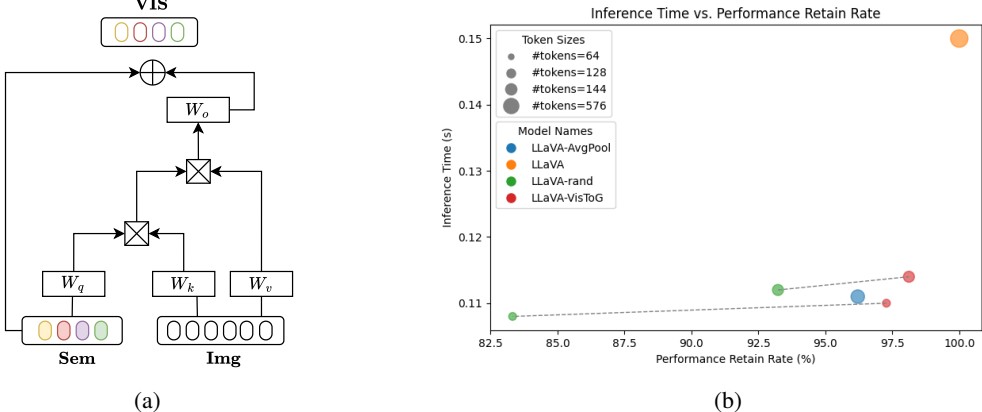

(a)          (b)

Figure 2: (a) Strucutre of the groupinglayer. (b) Comparison of Inference time and Average Performance between different models.

## 3.3 ISOLATED ATTENTION

To fully leverage the potential of the pre-trained vision encoder, it's crucial to mitigate the impact of the newly introduced semantic tokens, denoted as **Sem**, on the original image segment tokens. This ensures that the integrity of the original image representations remains intact. To achieve this, we implement a technique called isolated attention, which prevents the original image segment tokens from interacting directly with the newly added semantic tokens, thereby preserving their original characteristics.

More specifically, let the attention mask $M$ be a matrix defined as $M \in \mathbb{R}^{(M+N)\times(M+N)}$, where each element $M_{i,j}$ represents whether token $i$ can attend to token $j$. If $M_{i,j} = \texttt{True}$, token $i$ is allowed to attend to token $j$, and if $M_{i,j} = \texttt{False}$, attention is blocked between the two tokens. The isolated attention mechanism is then formally defined as follows:

$$
M_{ij} = \begin{cases} \texttt{False}, & \text{if } i \in \mathbf{Img} \text{ and } j \in \mathbf{Sem}, \\ \texttt{True}, & \text{otherwise.} \end{cases} \tag{5}
$$

This attention mask prevents the original image tokens **Img** from attending to the semantic tokens **Sem**, while allowing other interactions to proceed normally. Consequently, this design ensures that the output of the original image segment tokens remains unchanged despite the addition of the semantic tokens. By isolating the attention in this way, the original image tokens are preserved as they were before the introduction of the semantic tokens, maintaining the image's core representation.

On the other hand, the semantic tokens **Sem** can still learn to aggregate the image features derived from the pre-trained visual encoder into semantically meaningful regions that align with the provided instruction or task. This approach allows the semantic tokens to specialize in creating groupings of image features without disrupting the original token representations.

We further conduct a series of ablation studies on the attention mask configuration to validate the effectiveness of this isolated attention mechanism compared to using full attention, as detailed in Sec.4. These experiments demonstrate the importance of this isolation in retaining the fidelity of the image representations while allowing the semantic tokens to perform their intended role.

## 3.4 INSTRUCTION-AWARE VISUAL TOKEN GROUPING

Let $\phi$ denote the Large Language Model (LLM), $W$ denote the lightweight visual connector, which typically takes the form of a multi-layer perceptron (MLP), and $G$ represent the grouping layer. These components together form the foundation of our method for visual token grouping within an LLM-based framework. To effectively train the proposed VisToGto perform this task, we carefully design a two-stage training pipeline that ensures robust feature alignment and instruction-aware token grouping.

**Stage 1: Pre-training for Feature Alignment**

The goal of the first stage is to align the image features with the LLM by training an image tokenizer that maps visual inputs into a form compatible with the LLM. Since the pre-training phase operates on an image-caption dataset, the grouping mechanism is not yet emphasized during this stage, as the focus is primarily on aligning the visual representation with the caption, which serves as the natural language supervision. Given that the grouping mechanism is better suited to be learned from more specific instructions rather than general captions, we refrain from incorporating the grouping layer during this phase. Consequently, the pre-training phase follows the same setup as LLaVA Liu et al. (2024), where only the image-caption pairs are used to train the image tokenizer.

The key insight here is that, by not introducing the grouping mechanism prematurely, we allow the model to establish a strong foundation in feature alignment between images and language. In this stage, the only trainable parameter is the visual connector $W$, i.e., $\Theta = W$. This ensures that the visual features extracted from images are properly aligned with the LLM, setting the stage for further fine-tuning in the next phase.

**Stage 2: Visual Instruction Tuning**

Once the image tokenizer has been pre-trained and the visual features are aligned with the language model, we move on to the second stage, which focuses on fine-tuning the model for instruction-aware visual token grouping. At this point, we incorporate the grouping block $G$ along with semantic tokens into the Vision Transformer architecture to enable effective visual token grouping based on specific instructions.

In this stage, we freeze the pre-trained weights of the visual encoder to retain the feature alignment achieved in Stage 1. However, we continue updating the parameters of the lightweight visual connector $W$, the grouping layer $G$, and the LLM $\phi$. This means that the trainable parameters during Stage 2 are $\Theta = \phi, W, G$. Importantly, by introducing the grouping mechanism during instruction-tuning, we ensure that the grouping layer becomes instruction-aware, meaning that the grouping process is directly influenced by the specific instructions provided to the model.

The critical advantage of this approach is that the gradient of the instructions can flow back to the grouping layer, allowing it to learn how to group visual tokens based on the semantics of the instructions rather than general captions. This enables the `VisToG` to achieve a higher level of adaptability and precision when handling visual tasks that require instruction-specific groupings.

## 4 EXPERIMENTS

### 4.1 DATASETS

For fair comparison, we conduct experiments on the same datasets as introduced in Liu et al. (2024), which is ∼558K image-text pair for visual connector pre-training and ∼665K instruct-following data for visual instruct-tuning. Since a number of image links in the dataset of the instruction tuning stage have expired, compared to the original setting (∼665K), only ∼624K data are available. The performance of LLaVA1.5 reported in our analysis is reproduced by ourselves to ensure a fair comparison under the same experimental environment and dataset setting. For downstream tasks, we evaluate our model on GQA Hudson & Manning (2019), TextVQA Singh et al. (2019), POPE Li et al. (2023b), MMBench Liu et al. (2023), ScienceQA Lu et al. (2022), MME Fu et al. (2023)

### 4.2 BASELINES

We include results of BLIP-2 with Vicuna-13B as LLM backbone, InstructBLIP with Vicuna-7B and Vicuna-13B as LLM backbone, Qwen-VL/Qwen-VL-Chat with Qwen-7B as backbone. These models all adopt Q-Former to conduct the visual token abstraction and hence have a smaller number of image tokens compared with LLaVA (576 tokens). DeCo uses 2D adaptive average pooling to down-sample the visual tokens at the patch level and hence reduce the number of visual tokens to 144. C-Abstractor and D-Abstractor uses convolutional block and deformable attention block to conduct the visual token abstract, also resulting in 144 visual tokens. VoCo-LLaMA compresses the vision tokens using the LLMs as introduced in Sec.2. Here we include the results of 128 tokens for comparison. We also experiment on a very simple yet effective baseline. During inference, we randomly drop the vision tokens from $M$ to $M'$ before feeding into the visual connector. This method is denoted as LLaVA-rand.

### 4.3 MAIN RESULTS

We condcut all experiments on 8×NVIDIA100-40G and the training configurations are identical to that of LLaVA Liu et al. (2024).

From Tab.1 we can identify that LLaVA-rand serves as a considerable baseline. For example, in the 144 image token setting, it beats DeCo, C-Abstractor and D-Abstractor on GQA dataset and beats DeCo and D-Abstractor on MME dataset. In Fig. 3, we visualize how randomly selected image tokens of LLaVA-rand influence the output of visual question-answering tasks. On the left, we demonstrate that with randomly sampled $N = 64$ image tokens, the MLLM can achieve similar performance compared with LLaVA, which uses $N = 576$ image tokens. This highlights the redundancy inherent in the image tokens and establishes LLaVA-rand as a viable baseline, as further corroborated by the results in Tab.1. As long as the sampled image tokens include those representing critical segments of the image, the performance can be nearly equivalent to that of the original

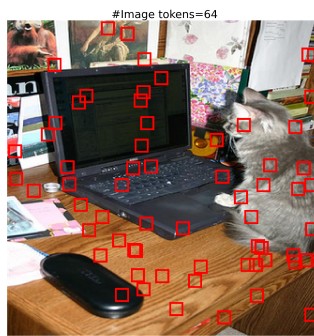 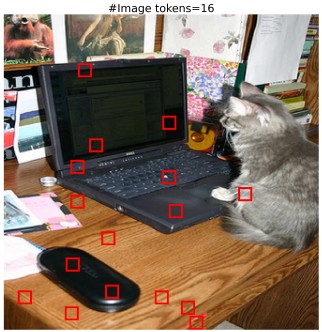 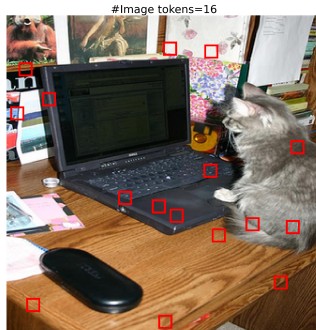

(left) The main focus of the image is a cat sitting in front of a laptop computer
(mid) The main focus of the image is a desk with a computer, which is located in a home office
(right) The main focus of the image is a cat on a desk in front of a computer.

Figure 3: Visualization of the image tokens selected of the LLaVA-rand. The instruction is "What is the main focus of the image?". Response from LLaVA: "The main focus of the image is a cat sitting on a desk in front of a laptop computer".

Table 1: Performance Comparison with leading methods. `VisToG` groups the visual tokens into 128 tokens and 64 tokens while achieving highly competitive performance compared with LLaVA. The results of `VisToG` are highlighted with purple.

| Method | LLM | Res. | #Tokens | IT | GQA | SQA | VQA$^\text{T}$ | POPE | MME | MMB |
|---|---|---|---|---|---|---|---|---|---|---|
| BLIP-2 | Vicuna-13B | 224 | 32 | 129M | - | 61 | 42.5 | 85.3 | 1293.8 | - |
| InstructBLIP | Vicuna-7B | 224 | 64 | 1.2M | - | 60.5 | 50.1 | - | - | 36 |
| InstructBLIP | Vicuna-13B | 224 | 64 | 129M | 49.5 | 63.1 | 50.7 | 78.9 | 1212.8 | - |
| Qwen-VL | Qwen-7B | 448 | 256 | 1.4B | 59.3 | 67.1 | 63.8 | - | - | 38.2 |
| Qwen-VL-Chat | Qwen-7B | 448 | 256 | 1.4B | 57.5 | 68.2 | 61.5 | - | 1487.5 | 60.6 |
| VoCo-LLaMA | Vicuna-7B | 336 | 128 | 665K | 59.8 | - | - | - | - | 61.0 |
| DeCo | Vicuna-7B | 336 | 144 | 665K | 54.1 | - | 56.2 | 85.9 | 1373.4 | - |
| C-Abstractor | Vicuna-7B | 336 | 144 | 665K | 52.6 | - | 55.9 | 84.5 | 1411.8 | - |
| D-Abstractor | Vicuna-7B | 336 | 144 | 665K | 53.1 | - | 55.1 | 84.6 | 1313.2 | - |
| LLaVA-1.5 | Vicuna-7B | 336 | 576 | 624K | 62.7 | 70.5 | 57.3 | 86.2 | 1452.0 | 64.3 |
| LLaVA-1.5-rand | Vicuna-7B | 336 | 144 | 624K | 57.3 | 70.5 | 50.4 | 79.5 | 1377.0 | 59.8 |
| LLaVA-1.5 + `VisToG` | Vicuna-7B | 336 | 128 | 624K | 61.4 | 70.1 | 54.5 | 85.5 | 1421.2 | 63.8 |
| LLaVA-1.5 + `VisToG` | Vicuna-7B | 336 | 64 | 624K | 60.9 | 70.9 | 52.5 | 85.7 | 1403.7 | 63.2 |

model. This nature also leads to the high variance of the performance of LLaVA-rand. In the middle, we demonstrate that with randomly sampled $N = 16$ image tokens, the model fails to recognize the presence of a cat. This failure is attributed to the insufficient number of tokens sampled from the cat's region, underscoring the importance of adequately covering meaningful semantic segments of the image using a limited budget. On the right, with the same number of visual tokens, we manually adjust the distribution of the sampled visual tokens using prior knowledge of the image. By ensuring the tokens are evenly allocated to each significant semantic object within the image, the model can recognize the cat once again. This adjustment demonstrates the importance of token distribution and allocation in achieving better performance, even with a limited number of tokens. More visualization can be found in the Appendix. This motivates our method, if we can automatically sample enough tokens for each important semantic group without a human interface, even with a small number of image tokens we can get considerable results.

We also include the average Performance Retain Rate in Tab. 2. Suppose the downstream datasets are defined by $D = \{D_i\}_{i=1}^{|D|}$, and let the performance of LLaVA with full image tokens on $D_i$ be denoted by $b_i$. For each model $j$, let its performance on $D_i$ be denoted by $p_i^{(j)}$. Then the average Performance Retain Rate (PRT) is defined by $PRT^{(j)} = \frac{1}{|D|} \sum \frac{p_i^{(j)}}{b_i}$.

Table 2: Comparison between average Performance Retain Rate (PRT,%) across GQA, TextVQA, POPE, and MME between different methods. The best results are marked as **bold**

| Method | #Tokens | PRT (%) |
|---|---|---|
| LLaVA-rand | 144 | 96.0 |
| DeCO | 144 | 94.7 |
| C-Abstractor | 144 | 92.9 |
| D-Abstractor | 144 | 93.0 |
| VisToG | 128 | **97.4** |

## 4.4 INFERENCE EFFICIENCY

To verify the efficiency of VisToG, we conduct experiments on calculating the inference time of different methods. We include the performance of LLaVA-rand as defined before and also re-implement the Adaptive Average Pooling as used in Yao et al. (2024), denoted as LLaVA-AvgPool. We calculate the per-sample inference time on different downstream datasets. Let $\{D_i\}_{i=1}^k$ denote the downstream datasets that we aim to infer. For model $j$, let $t_{ij}$ denote the total inference time on $D_i$, then average inference time $T_j = \frac{1}{k} \sum_{i=1}^{k} \frac{t_{ij}}{|D_i|}$. Since the performance of downstream datasets is in different scales, we calculate the Performance Retain Rate (PRT, %), which is defined as the ratio as compared to the baseline LLaVA that uses all image tokens. As shown in Fig. 2b, the baseline LLaVA that uses all 576 image tokens has an average inference time of 0.15s. As for LLaVA-VisToG, it has a significantly higher PRT compared to the LLaVA-rand counterparts while only sacrificing a negligible inference time. While the performance of LLaVA-AvgPool falls between them. All the inference experiments are conducted on a single NVIDIA L40S and are averaged across three runs.

## 4.5 ABLATION STUDY

### 4.5.1 DIFFERENT ATTENTION MASKS

To verify the effectiveness of the isolated attention, we conduct experiments using the standard full attention of the vision transformer across six downstream datasets. We report the average performance retain rate defined before as the performance metric. As can be seen in Fig.4, we include results of both #tokens=64 and #tokens=128. For both token counts, isolated attention consistently outperforms or matches the performance of full attention across all datasets. Specifically, for #tokens = 64, isolated attention achieves slightly higher performance in GQA, SQA, POPE, MME, and MMB, while showing a notable improvement in TextVQA as well. Similarly, for #tokens = 128, isolated attention continues to demonstrate superior performance in GQA, SQA, POPE, MME, and MMB, with a significant lead in TextVQA. These results suggest that isolated attention is more effective than full attention, particularly in scenarios with varying token counts, making it a more efficient mechanism for multi-modal large language models. The consistent performance advantage of isolated attention across different datasets and token counts underscores its robustness and potential for enhancing the efficiency of multi-modal models.

### 4.5.2 NUMBER OF IMAGE TOKENS

As in Fig. 5, we conduct experiments across varying visual tokens and show the result of VisToG, LLaVA-rand and LLaVA-AvgPool on GQA and POPE. As observed in subfigure (a), VisToGconsistently outperforms LLAVA-rand and LLAVA-AvgPool across all token counts, maintaining relatively high performance even as the number of tokens decreases. LLAVA-AvgPool shows a slight decline in performance but remains relatively stable, whereas LLAVA-rand exhibits a sharp drop when the token count falls below 64. Fig 5 (b) shows a similar trend on the GQA dataset, with VisToG again achieving the highest performance across all token counts. LLAVA-AvgPool maintains a moderate performance level, while LLAVA-rand experiences a significant degradation in performance as the token count decreases, particularly when reduced to 32 tokens or fewer. These results suggest that VisToG is the most robust method in handling reduced token counts, while

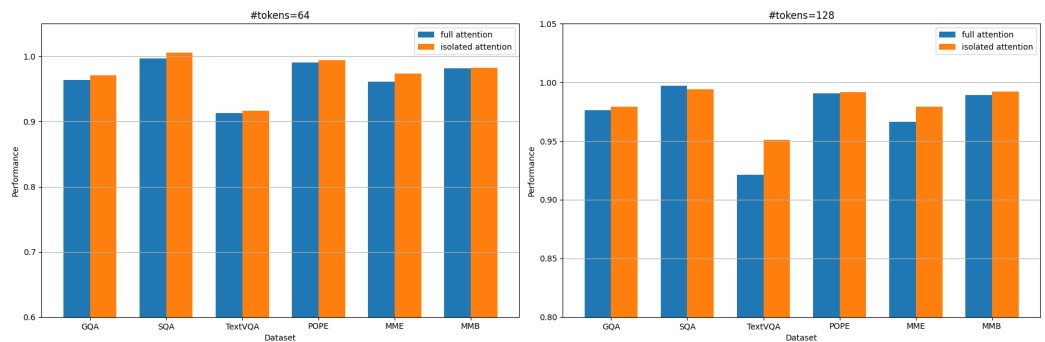

Figure 4: Performance Comparison between standard attention and isolated attention. The numbers are the relative performance compared to the baseline.

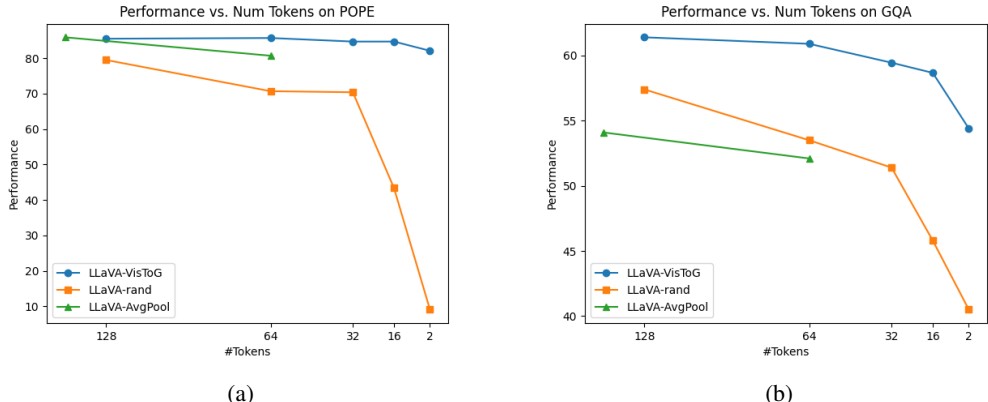

(a)                 (b)

Figure 5: (a) Ablation on the number of image tokens on the POPE dataset. (b) Ablation on the number of image tokens on GQA dataset.

LLAVA-rand is highly sensitive to token reduction, indicating the randomness causes loss of visual information when the visual tokens are very limited.

## 5 CONCLUSION

In this paper, we have introduced `VisToG`, a novel grouping mechanism designed to address the substantial computational costs associated with Multi-modal Large Language Models (MLLMs). By leveraging pre-trained vision encoders to group similar image segments without the need for additional segmentation masks, `VisToG` effectively reduces inference costs. Our approach utilizes isolated attention to identify and eliminate redundant visual tokens, significantly decreasing computational demands. Extensive experiments validate the efficacy of `VisToG`, demonstrating that it maintains 98.1% of the original performance while achieving a reduction of over 27% in inference time. This advancement enhances the efficiency of MLLMs and provides insights on training larger MLLMs with minimal image token redundancies.

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

## A  APPENDIX

You may include other additional sections here.

