# OpenReview forum: "Efficient Multi-modal Large Language Models via Visual Token Grouping"
_ICLR.cc/2025/Conference — ICLR 2025 Conference Withdrawn Submission_

### Official Review · Reviewer_eHc4 · 2024-10-24

**Soundness:** 3
**Presentation:** 3
**Contribution:** 3
**Rating:** 6
**Confidence:** 4

**Summary:**

This paper proposes a method for compressing visual tokens in the multimodal large model by adding additional learnable tokens in the vision transformer. It designs a attention mask of the vision transformer to keep original output of vision transformer and compresses visual tokens into learnable tokens using a method similar to cross attention. This approach compresses the size of the visual tokens while maintaining a certain level of model performance reduction.

**Strengths:**

The method has a certain level of innovation.

**Weaknesses:**

More comparative experiments on visual feature compression under the same experimental settings are needed.

**Questions:**

Have you tried using a visual token compression method similar to QwenVL on LLava1.5?

---

### Official Review · Reviewer_HKkH · 2024-10-29

**Soundness:** 3
**Presentation:** 2
**Contribution:** 2
**Rating:** 5
**Confidence:** 5

**Summary:**

This paper introduces VisToG, a vision token compression method for efficient inference in large vision-language models. The basic idea is to group the similar vision tokens with the token similarity computation, and leverage the vision encoder to initialize the group tokens. The training configurations adopt the LLaVA-v1.5 style. The experiments cover the comparison with previous methods on vision token compression and some ablation studies, where VisToG achieves promising lower inference cost.

**Strengths:**

1. The overall design is technically sound, which can be easily implemented.
2. This paper offers a clear background on why we need vision token compression in vision-language models.

**Weaknesses:**

1. The overall pipeline is highly similar with Q-Former based vision token compression methods. After going though the paper, I feel the only differences are two particular designs:
- The first one is encouraging the vision encoder to initialize the query tokens (i.e., the group tokens in this paper) by learning some tokens to abstract the vision information in patch tokens of the vision encoder;
- The second one is using Gumbel-SoftMax based operation to calculate the Q-K similarity, resulting in the one-hot selection on vision tokens rather than the weighted sum of vision tokens. Also, it only requires a single grouping layer.

However, the motivation of the above two designs remains unclear. I cannot find any ablation results in the paper to validate the effectiveness of these designs. Especially, I am wondering the model performance if using common cross attention rather than Gumbel-SoftMax based vision token selection. To be honest, it is really hard to comprehend the motivation of the second design.

2. As shown in Table 1, the proposed method does not appear fully comparable to the vanilla LLaVA-v1.5 baseline, especially with a notable drop of approximately 5 points on TextVQA. In my experiences, a 5% decline in multi-modal benchmark scores often signals a disproportionately larger impact on a model's multi-modal capabilities. Therefore, it is unconvincing to claim that the model ‘retains xx% performance’ just based on the percentage drop in benchmark scores.

3. And, It is still hard to judge the superiority of VisToG compared with Q-Former. Maybe the authors could conduct a fair comparison on the LLaVA-v1.5 backbone, with the same data and training settings. That will be much better.

4. Please show the full results of table 2 rather than the average results.

5. Not good presentation with some typos like "8×NVIDIA100-40G" in Line 314.

**Questions:**

The biggest concern should be the motivation of the two designs, and the current experiments cannot support to prove the effectiveness of  them. I think the authors should well figure out these points before claiming the necessity of introducing this method.

**Details Of Ethics Concerns:**

I don't think there is any necessity for ethics reviews for this paper.

---

### Official Review · Reviewer_ZMBg · 2024-11-02

**Soundness:** 2
**Presentation:** 2
**Contribution:** 2
**Rating:** 3
**Confidence:** 4

**Summary:**

To reduce the computational costs associated with MLLMs, the authors propose a method that leverages pre-trained vision encoders to group similar image segments into semantically related concepts without the need for segmentation masks. Besides, the method employs isolated attention to preserve the integrity of original image representations while allowing semantic tokens to aggregate image features into meaningful regions.

**Strengths:**

* Despite reducing computational demands, the method maintains 98.1% of the original performance, indicating that it is highly efficient without compromising on accuracy.
* By reducing the number of visual tokens processed by the model, the method is more scalable and flexible than original MLLMs.

**Weaknesses:**

* My main concern is the novelty of the proposed method. The use of clustering algorithms or q-formers to reduce the number of vision tokens fed into LLMs has been examined in several previous works, including Chat-UniVi [1]. Additionally, the concept of isolated attention is not novel. I recommend that the authors provide a more in-depth analysis of the proposed method to strengthen the paper.
* To demonstrate the generalizability of the proposed method, I suggest that the authors validate it across a broader range of MLLM architectures or base LLMs.

[1] Jin, P., Takanobu, R., Zhang, W., Cao, X., & Yuan, L. (2024). Chat-univi: Unified visual representation empowers large language models with image and video understanding. In Proceedings of the IEEE/CVF Conference on Computer Vision and Pattern Recognition (pp. 13700-13710).

**Questions:**

* LLaMA-VID [1] can encode images and videos using only two tokens, whereas the proposed method requires a minimum of 64 tokens and does not support video input. What advantages does the proposed method offer over LLaMA-VID?

[1] Li, Y., Wang, C., & Jia, J. (2025). Llama-vid: An image is worth 2 tokens in large language models. In European Conference on Computer Vision (pp. 323-340). Springer, Cham.

---

### Note · Authors · 2024-11-15

I have read and agree with the venue's withdrawal policy on behalf of myself and my co-authors.